# The Costs of Living with Floods in the Jamuna Floodplain in Bangladesh

**Md Ruknul Ferdous** [1,2,*] , **Anna Wesselink** [1] , **Luigia Brandimarte** [3] , **Kymo Slager** [4] ,
**Margreet Zwarteveen** [1,2] and **Giuliano Di Baldassarre** [1,5,6]

[1] Department of Integrated Water Systems and Governance, IHE Delft Institute for Water Education,
2611 AX Delft, The Netherlands; a.wesselink@un-ihe.org (A.W.); m.zwarteveen@un-ihe.org (M.Z.);
giuliano.dibaldassarre@geo.uu.se (G.D.B.)

[2] Faculty of Social and Behavioural Sciences, University of Amsterdam,
1012 WX Amsterdam, The Netherlands

[3] Department of Sustainable Development, Environmental Science and Engineering, KTH,
SE-100 44 Stockholm, Sweden; luigia.brandimarte@abe.kth.se

[4] Deltares, 2600 MH Delft, The Netherlands; Kymo.Slager@deltares.nl

[5] Department of Earth Sciences, Uppsala University, SE-75236 Uppsala, Sweden

[6] Centre of Natural Hazards and Disaster Science, CNDS, SE-75236 Uppsala, Sweden

* Correspondence: r.ferdous@un-ihe.org

**Abstract:** Bangladeshi people use multiple strategies to live with flooding events and associated riverbank erosion. They relocate, evacuate their homes temporarily, change cropping patterns, and supplement their income from migrating household members. In this way, they can reduce the negative impact of floods on their livelihoods. However, these societal responses also have negative outcomes, such as impoverishment. This research collects quantitative household data and analyzes changes of livelihood conditions over recent decades in a large floodplain area in north-west Bangladesh. It is found that while residents cope with flooding events, they do not achieve successful adaptation. With every flooding, people lose income and assets, which they can only partially recover. As such, they are getting poorer, and therefore less able to make structural adjustments that would allow adaptation in the longer term.

**Keywords:** flooding; erosion; coping; adaptation; Jamuna River; Bangladesh

## 1. Introduction

The image of Bangladesh as a country that is adapting well stems from its long history of living with floods. Bangladesh is a riverine country and one of the most flood-prone countries in the world [1–3]. The country is in the largest delta of the world, which developed, and is continuously changing, through processes of sedimentation and erosion by the three mighty rivers Ganges, Brahmaputra, and Meghna. Bangladesh is also a very densely populated country with most people depending on agriculture and fisheries for their livelihoods [4]. Most of the population lives in floodplain areas, with varying degrees of exposure to riverine flooding and riverbank erosion [5]. In Bangladesh, floods can be both resources and hazards. In the monsoon season, 25–30% of the floodplain area is typically inundated by the so-called normal floods (*borsha*). This is considered beneficial as flooding increases the fertility of the land by depositing fresh silt on the soil and replenishing soil moisture without doing too much damage to property or disruption to traffic and commerce. However, an abnormal flood (*bonna*) is considered a hazard because of the damage it does to crops and properties, and the threat to human lives [6,7]. Riverbank erosion is another hazard in times of flood and occurs almost in every year, whether the flood is normal or abnormal. Abnormal floods and riverbank erosion not

only cause substantial damage to inhabitants' agricultural lands, crops, and properties, they also have wider socio-economic consequences, for example a decline in agricultural production [8] and a steady flow of migrants to urban areas [9–11]. Each year several thousands of people become temporarily or permanently homeless and/or landless and take refuge to nearby embankments or on neighbors' or relatives' land. The extremely poor people who live on the islands in the wide (up to 16 km) rivers (called chars) are most exposed to and affected by flood hazards and riverbank erosion. In the 1988 flood, more than 45 million people were displaced and over 3000 died [12,13]. Flood mortality rates have declined since 1988 because of better institutional support during floods [14]. Yet, the socio-economic disruption caused by flooding events is still considerable [3,15].

"Living in a low-lying and densely populated country on the front line of climate change, Bangladeshis are taking a lead in adapting to rising temperatures and campaigning to limit climate change. Bangladeshis will keep their heads above water, but at huge costs" [1]. This back cover summary of the recent book "*Bangladesh Confronts Climate Change: Keeping Our Heads Above Water*" [1] portrays a common, positive image of Bangladesh as being able to respond successfully, or adapt, to climate change (see also [12,16–19]). However, it also indicates that this response comes with huge costs. Floodplain inhabitants use a wide range of tried-and-tested strategies to cope with the conditions during and after the flooding. However, repeated exposure to flooding events often means impoverishment.

Only a few authors have paid sufficient attention to this effect, which can be seen as a sign of maladaptation [20]. Most research done on people's abilities to cope or adapt is based on qualitative statements from floodplain inhabitants, without looking at the cumulative effect of floods over a longer time period (i.e., across decades). As time series of quantitative data to substantiate these claims are not available from secondary sources, estimates of flood damage often rely on unverifiable proxies [21]. More specifically, various studies assert that people are getting poorer because of recurring floods [12,22,23], but most of these do not provide quantitative substantiation. One of the processes that may lead to impoverishment is diminished agricultural production in times of disastrous floods [8]. Rural people are most affected by these floods; they are also suffering from the persistent effects of labor market disruption and income deficiency [24]. Moreover, they must spend a large portion of their income on food and repairing or constructing new houses, as their old houses are often ruined by floods and riverbank erosion. Their savings are reduced to zero and the other necessary expenses to survive are only increasing their burden [25].

A few studies quantify the average household losses due to severe floods and the associated riverbank erosion. Chowdhury [26] presents a figure of 47 and 88 USD/year for the years 1986–1987, in two locations. Ten years later Thompson and Tod [27] estimate losses to homesteads to be 132 and 190 USD/year for the 1988 and 1991 floods, again in two locations. The most complete figures are given by Paul and Routray [3], who estimate income losses at 82 and 108 USD/year and asset losses 191 and 257 during the 2005 flood, again in two locations. Combined with direct loss of assets in floods, and loss of the key productive asset (land) to erosion, severe floods in the chars make vulnerable households and the community poorer [27]. Finally, they are falling into debt and impoverishment [12,22,23,28]. Further indebtedness leads to even more drastic regimes of "eating simpler food", increased malnutrition, increased levels of disease, increased numbers of wives left to feed children alone while their husbands are temporarily migrated for work, increased sales of remaining assets, increased rural landlessness, increased migration to the cities, and increased vulnerability to future floods.

Moreover, only a few authors investigate the distributional effects of flood impacts. Paul and Routray [3] show how the adoption of a particular set of adjustment strategies depends on people's socio-economic circumstances, such as education, income, and occupation. Islam et al. [29] look at adjustment strategies against flood and riverbank erosion of the char inhabitants of the Jamuna River. They also find that household's ability to cope with flood and river erosion depends on people's socio-economic and environmental conditions. Hutton and Haque [28] assert that impoverishment and marginalization in part reflect inequitable access to land and other resources. The likelihood of

impoverishment of the household is further increased not only by social and demographic factors (including gender, education, health and age), but also by underlying economic and social relationships which can increase human vulnerability to risk. Indra [30] investigates the forced migration due to riverbank erosion (and hence loss of land) in the Jamuna floodplain, and concludes that most people displaced by riverbank erosion are already poor and disempowered before being uprooted by the shifting channels of the Jamuna River.

This paper critically assesses the idea that Bangladeshis have a good adaptive capacity to changing environmental conditions. To this end, it examines the socio-economic effects of repeated floods and riverbank erosion on rural households, and assesses whether and how different households deal with this flooding. The aim is to uncover whether and how households succeed in adapting to flood conditions, as opposed to those who are merely coping. As defined by Berman et al. [31] coping refers to immediate responses to events, while adapting prepares households for expected future events. These definitions are extended as follows: coping includes long-term adjustments where the net result can be a decline in socio-economic conditions, while adapting means that people continue to live as well, or better than before, and are as well, or better than before, able to cope with future events. Both coping and adapting require the mobilization of a variety of technologies and social or institutional changes, which is defined as adjustments. The actions and strategies for adjustment may be the same, but have a different result: coping or adapting. In general, coping strategies are those that are possible within the current settings, and adaptation strategies are more likely to involve more fundamental changes in the type of livelihood activity or location [32].

## 2. Case Study

To understand how Bangladeshi cope with flooding, this study focuses on the floodplain area along a 30 km reach of the Jamuna River in the north of Bangladesh. The study area includes parts of Gaibandha district and parts of Jamalpur district (Figure 1). The total surface is about 500 km$^2$ and the total population is approximately 0.36 million people [4,33]. This is a unique case study since three adjacent areas are inhabited by people with similar socio-economic conditions, but different levels of flood protection and exposure to riverbank erosion (Figure 1).

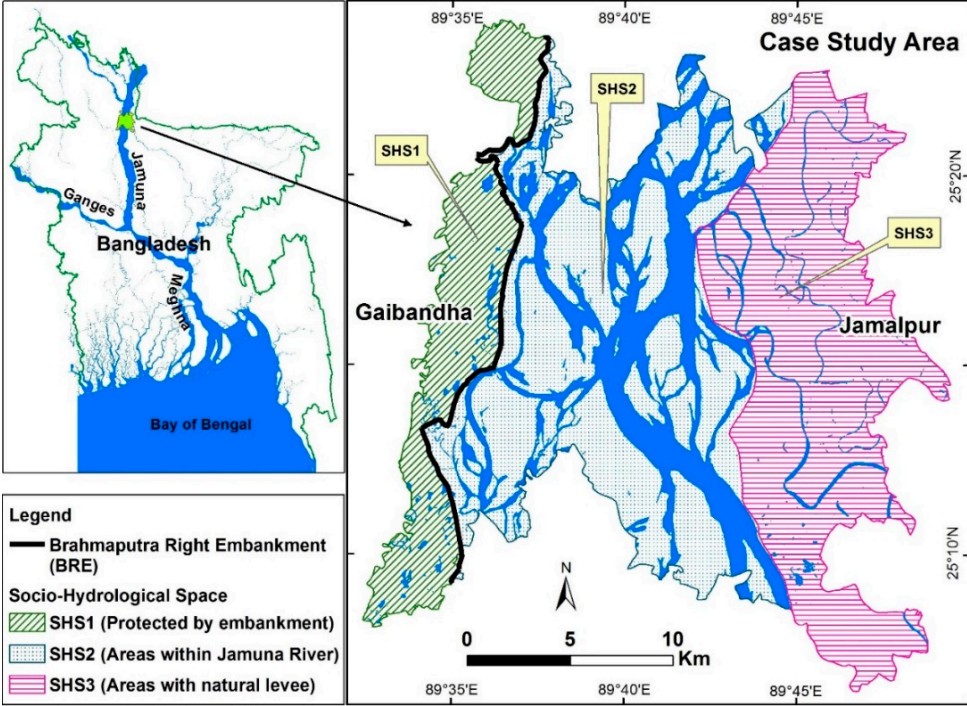

**Figure 1.** Bangladesh map with study area [34].

To investigate the effect of floods and riverbank erosion on livelihood conditions, the area is divided into three socio-hydrological spaces [34]. The socio-economic situations are also classified based on income levels and assets (Section 2.2).

### 2.1. Socio-Hydrological Spaces

Figure 1 shows the braided riverbed, which includes many inhabited river islands (*chars*) that get flooded with varying frequency (some every year, some only with severe floods). The area to the west of the river is protected by a human-made embankment, while the area to the east of the river only has a natural levee deposited by the river. These different physical conditions create different flood protection levels, which in turn give rise to different socio-economic responses and conditions. To reflect this socio-spatial differentiation, the authors have elsewhere explained and motivated the classification of the study area into three socio-hydrological spaces, abbreviated to socio-hydrological spaces (SHS) [34] (Figure 1). These spaces are briefly described below.

SHS1 is an area protected by an artificial levee (the Brahmaputra Right Embankment, BRE) constructed in the 1960s to limit flooding and increase the agricultural production of that area. This area is protected from regular annual flooding, but the maintenance of the BRE in the study area has been sporadic, and breaches occur during abnormal floods, causing catastrophic floods and damages [35]. Also, two small rivers Ghahot and Alai inundate some parts of the area, and other parts are frequently inundated with excess rainwater, due to their low elevation and limited drainage capacity. The total area is about 74 km$^2$ with a population of approximately 111,000 [4].

SHS2 is the floodplain within the embankment on the west bank and the natural levee on the east bank. In SHS2, flooding occurs most frequently, essentially every year. This area includes extensive chars (river islands), where multiple channels crisscross within the outer boundary of the riverbed. These chars can shift in space due to continuous processes of deposition and erosion of river sediments. The stability of the chars depends on their age. Some older chars have higher elevations than the areas in SHS1 and remain dry during flood conditions. If a new char develops, homeless people analyze the stability of the new char and after 2–3 years start activities such as farming or living on the char. When the age of the char exceeds about 20 years it is called a stable char [36]. Nevertheless, all chars can shift during the flooding events. The total area is about 246 km$^2$ with a population of approximately 104,000 [4].

SHS3 is the area on the east bank which is without any man-made flood protection. Flooding occurs here more frequently than SHS1, around once in two years [34]. This area is sometimes flooded by adjacent small rivers, in this case the Old Brahmaputra and Jinjira, as well as by excess rainfall. Riverbank erosion is also prominent in this area. Inhabitants take the initiative to build small spurs and bank protection made from bamboo and wood, to try to stop erosion. However, while these encourage sedimentation at a local scale, they are not sufficient to stop large scale erosion. The total area is about 174 km$^2$ with a population of approximately 146,000 [4,33].

Decadal data from 1961 to 2011 show that the population density has increased from 300 persons/km$^2$ to about 800 persons/km$^2$ over the whole study area [4,33]. However, population density in the three spaces differs substantially. In SHS1 it has always been higher than in the SHS2 and SHS3 (Figure 2). The density in SHS1 has increased from 600 to 1500 person/km$^2$, a rate of 18 persons/km$^2$/year. In SHS2 population density has increased from 200 to 400 person/km$^2$, a rate of 4 persons/km$^2$/year, and in SHS3 has increased from 300 to 800 person/km$^2$, a rate of 10 persons/km$^2$/year.

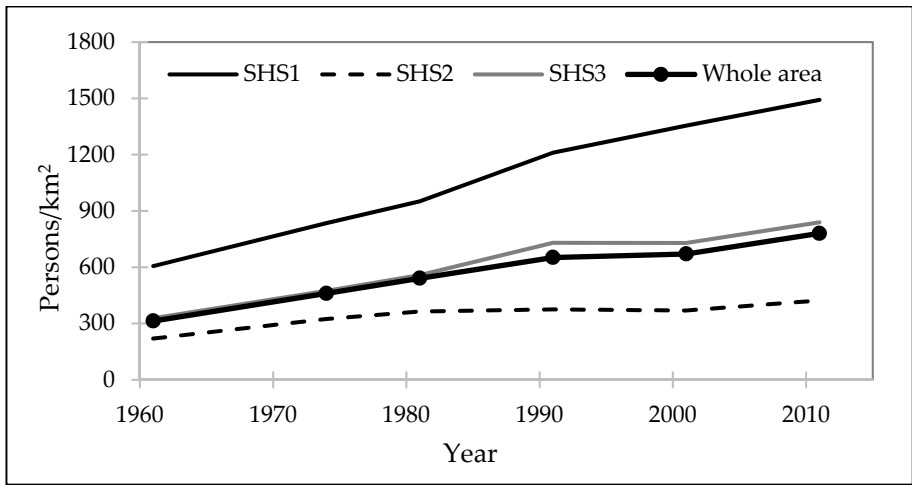

**Figure 2.** Population density of the study area in the period 1961–2011.

*2.2. Socio-Economic Conditions*

Data on socio-economic activities are sparse. The only historic economic data (from 1971) pertain to the number of commercial establishments developed for non-farming activities. These data are aggregated by districts that are larger than the SHS considered here. An establishment is defined an enterprise or part of an enterprise that is situated in a single location and in which only a single (non-ancillary) productive activity is carried out or in which the principal productive activity accounts for most of the value added [37,38]. Table 1 shows that the socio-economic situation in Gaibandha district was a little better than in Jamalpur district in 1971, but they were almost the same in 2013. The growth in the number of establishments changed dramatically around the year 2000 (Table 1). To get information about the company size and its impact on the local economy, this research uses the number of employees as a proxy. This information is only available for 2003 and 2013 (Table 1).

**Table 1.** Non-farm economic activities: number of establishments and employees [37,38].

| Year | Gaibandha District | | | Jamalpur District | | |
|---|---|---|---|---|---|---|
| | Establishments | Persons Engaged | Persons/ Establishment | Establishments | Persons Engaged | Persons/ Establishment |
| 1971 | 2675 | | | 1931 | | |
| 1989 | 10,540 | | | 9329 | | |
| 1999 | 33,651 | | | 32,304 | | |
| 2003 | 62,655 | 155,094 | 0.404 | 54,724 | 128,366 | 0.426 |
| 2009 | 121,495 | | | 127,012 | | |
| 2013 | 151,052 | 318,579 | 0.474 | 159,156 | 299,997 | 0.531 |

Table 1 seems to suggest that long-term economic trend in the area is upwards. Yet, agricultural farm sizes decreased in the study area since the 1960s (Figure 3). This is a potential sign of impoverishment. Figure 3 shows that the number of large farms decreased from 10% to only 1% during the period 1960s to 2016 and the number of landless households increased from 10% to about 37% during the same period.

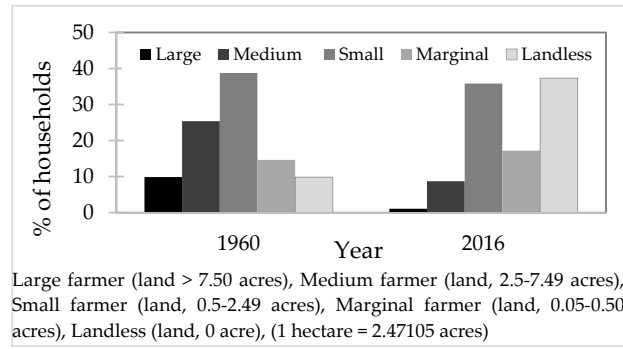

Large farmer (land > 7.50 acres), Medium farmer (land, 2.5-7.49 acres), Small farmer (land, 0.5-2.49 acres), Marginal farmer (land, 0.05-0.50 acres), Landless (land, 0 acre), (1 hectare = 2.47105 acres)

**Figure 3.** Number of non-farming establishments and change in agricultural farms in the study area.

In the study area, more than 80% of households rely principally on farm incomes, so agricultural income and assets are key indicators for socio-economic status. The existing Bangladesh Government classification of farm sizes only accounts for land holding size [39]. However, households have other assets (homesteads, equipment, cash or jewelry) that help them survive in case of flooding or riverbank erosion. Moreover, differences in average income affect how households can cope with such events. Data of this research include current annual income, annual expenditure and total wealth (see Section 3 for details about data collection). Using these data, the surveyed households are divided into five socio-economic classes that combine current wealth and income (Table 2). To increase the statistical significance of comparisons between classes, the class boundaries are chosen in such a way that each class contains an almost equal number of households.

**Table 2.** Classification of socio-economic status of the people in the study area.

| Socio-Economic Classes | Total Wealth and Yearly Income |
|---|---|
| Poor | Little wealth ≤ 5000 USD, Low income ≤ 750 USD |
| Moderate poor | Little wealth ≤ 5000 USD, Moderate income > 750 USD |
| Moderate | Moderate wealth 5000–30,000 USD, Moderate income ≤ 1000 USD |
| Moderate rich | Moderate wealth 5000–30,000 USD, Moderate income > 1000 USD |
| Rich | High wealth > 30,000 USD, Moderate income > 1000 USD |

By comparing the current distribution of socio-economic classes in the three SHS, see Section 2.1 and Figure 1), one can see remarkable differences (Figure 4a). In SHS1, 10% of the households are poor and 35% are rich, while in SHS2 it is just opposite. SHS3 takes an intermediate position, with 15% poor households and 25% rich. Using the Government farm size classification for comparison (Figure 4b), it is observed that more than 35% of households are landless in the study area and only 1% of households are large (Figure 4b). Most of the landless farmers are found in the SHS2 as they are experiencing much more land erosion than SHS1 and SHS3.

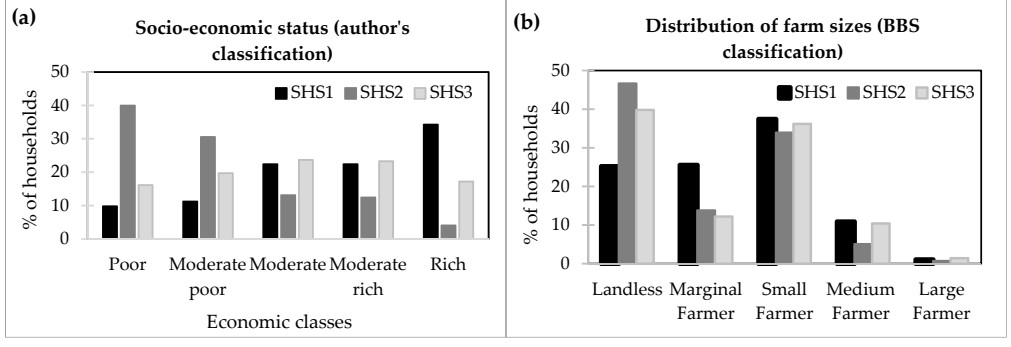

**Figure 4.** Current socio-economic status of the households in the study area by socio-hydrological space. (**a**) Socio-economic status (author's classification); (**b**) Distribution of farm sizes (BBS classification).

*2.3. Adjustment Strategies in the Study Area*

Inhabitants of the study area employ a range of strategies to adjust with flooding. These can be categorized into those that allow survival in times of flood and those that allow long-term living with floods and riverbank erosion.

To survive during flooding, people eat fewer meals, borrow money or take a loan, or sell their labor cheaply in advance [25]. If necessary they also sell their land, livestock, housing materials and other personal belongings, including jewelry and household goods [27]. In char areas (SHS2, Figure 1) the inhabitants only leave their homes when their lives are at risk. When high floods erode their land, they dismantle their houses and transport them to another char which is less (or not) affected by flooding and erosion [40]. If necessary, people take shelter on the embankments, with their livestock, in the hope that they might return in the near future to the re-emerged land, where they have property rights [15]. In most cases, these hopes are not fulfilled because it may take decades for land to re-appear [41]. Others move to a nearby relative's or friend's house or migrate temporarily to other districts looking for temporary work [9,30,42–45]. It is noted that main roads and railways are built on embankments to raise them above high flood levels. Rural roads and paths between settlements generally follow the highest land available and are usually also built on embankments to raise them above normal flood levels. This somewhat limits disruptions during floods, and provide an emergency refuge.

To structurally improve their chances of survival during a flood, inhabitants adjust their homes. In the char areas (SHS2), houses are built in a way that they can be easily dismantled [30,42]. In SHS1 and SHS3, homesteads are generally built on natural elevations, or on artificial earthen mounds, the height of which is determined by local experience of previous high flood levels [17]. The plinth of houses is further raised by digging earth from local depressions. Especially on the chars (SHS2) people also build platforms inside their homes to take shelter using bamboo, straw, water hyacinth, and banana stalks during abnormal flooding years [12].

To improve livelihoods, rural inhabitants have developed farming practices that are adjusted to the height, duration, and timing of normal floods, i.e., that commence and recede in time and attain normal height [6,45]. Rasid and Mallik [46] observe that the most common adjustment of rice cropping to the uncertainties of the flood regime is evident from the practice of intercropping two rice varieties, *aus* and *aman*, together. This measure ensures that at least flood-tolerant *aman* would be secured during an abnormal flood regime, even if the flood-vulnerable *aus* is lost or damaged. During a normal flood regime both *aus* and *aman* would succeed, often resulting in a bumper crop. Farmers have made a careful selection of the best adjusted varieties of rice over the centuries, to enable them to face floods [16], and they select other crops to sow off-season to fit the land elevation [12]. Other adjustment strategies include early harvesting of *aus* in case of excess or early flooding, planting older and taller seedlings on lands liable to repeated flooding, re-transplanting salvaged seedlings, protection of rice plants from water hyacinth by floating bamboo fences, and the post-flood cultivation of lentils, pulses, mustard, as well as winter vegetables and wheat [46].

While temporary migration can be a survival strategy (see above), migration can also be a long-term adjustment to increase household incomes and livelihood security [45]. Very few people move permanently to towns and cities [10,28,41]; the majority of flood affected people try to stay near their homes because kinship ties mean that local inhabitants help each other in crisis situations [2]. In that case, migration often concerns one or more family members who migrate, usually men. This can be on a seasonal basis, to other rural areas as agricultural laborer, or to urban areas as unskilled laborer or rickshaw driver, or more permanently to work in factories. Permanent migration out of the area of origin is rare, and mostly related to the loss of land due to riverbank erosion, while loss of livestock and crop failure more likely leads to temporary migration [10,30]. However, migration often comes at a cost: working conditions tend to be challenging and dangerous. People who are forced to migrate permanently from rural to urban areas often end up in slums with difficult living conditions.

### 3. Data and Methods

Primary data and secondary data were collected to explore how households in the study area adjust to regular flooding and riverbank erosion, as well as the long-term socio-economic effects of these adjustments.

Household surveys and focus group discussions (participants selected from the previously surveyed households) were performed during the dry seasons of 2015 and 2016. The principal set of primary data consists of 863 questionnaires dealing with several themes: general information (location of settlement and agricultural land, main occupation, age, income and expenditures, wealth and origin of the households), information on different flood experiences (depth of floods, frequency, duration, flood damages, effects on agricultural income and expenditures, other adjustment options such as migration) and experiences with river erosion (frequency, damages, migration, adjustment options etc.). For several of these aspects the respondents were asked to compile a historical record going back to 1960, which is why the households headed by older men or (occasionally) women were selected. This is a limitation of the study. Many old men were surveyed since they experienced many flooding events. Although other members of the family were typically present during the surveys, this choice may have introduced a selection bias. 12 focus group discussions were also set up in the study area to validate and contextualize survey data.

Since the respondents were asked to recollect their flood experiences going back to 1960, inaccuracies due to memory loss is one of the limitations of the household survey data. When asked to recollect major flood events and details of flood damages, respondents could easily remember events in the last two decades, but they were not so confident about the floods before 1980s. To handle this issue, the questionnaires were filled in in the presence of other family members, who helped the primary respondent to remember details about the past. The surveyors also used references to remarkable years, such as the year of independence of Bangladesh in 1971, or the construction period of embankments, to connect the respondents with years of major flood events.

To further check the reliability of the respondent's flood memory, the surveyed data are compared with both Government data and published journal articles. Based on flood duration, exposure, depth, and damage, the Flood Forecasting and Warning Centre (FFWC) of Bangladesh Water Development Board (BWDB) classifies flood events into three categories: normal, moderate, and severe. According to this classification, Bangladesh has experienced 9 severe floods since 1950s, namely in 1954, 1955, 1974, 1987, 1988, 1998, 2004, 2007 and 2017 [47]. However, according to household surveys a total of 33 major floods were experienced in the study area since the 1960s. A plausible reason for this difference is that the study area is very close to the Jamuna River, so households are facing huge damages almost every year. As such, almost all floods have severe consequences in the study area.

In the literature different lists of severe floods are presented. Hossain et al. [48] report major flood events in 1954, 1955, 1956, 1962, 1963, 1968, 1970, 1971, 1974, and 1984. Brammer [17] mentions that 16 major floods struck Bangladesh in the years between 1950 and 2000, with additional floods in 1977, 1980, 1987, 1988, 1998, and 2000. The respondents in the study could easily remember those flood events mentioned by Brammer [17] except the flood of 1977. According to Paul [16], Bangladesh has experienced 28 major floods in the past 42 years (1954–1996), of which 11 were classified as "devastating" and five as "most devastating". Summarizing all published flood information in the journal articles, 30 major floods were observed during 1962–2017. This flood count is very close to the numbers that the study collected from the household surveys. This suggest that the memory of the respondents is good enough to compile a record of historical flood events in the study area.

For the data on land loss due to riverbank erosion, all respondents were confident that they could remember the actual losses and the years they occurred. According to them, the loss of land is never forgotten. Respondents claimed to remember very well how much lands they had in 1960s, how much land had eroded since then, and in which year. They said that it is possible to recover from flooding, but that it is not possible to recover losses from riverbank erosion.

## 4. Results

Statistical analyses are performed for the whole study area, for each SHS (Figure 1), and for the different socio-economic classes (Table 2). This is done through statistical analysis of single variables and correlation between variables. ANOVA test (p < 0.05) and Chi-square tests are also performed to verify the significance of the outcomes. Statistical test summaries are given in the Electronic Supplementary Material (ESM).

### 4.1. Adjustment Strategies

Relocation means to settle permanently in another place with the whole household. Households in the study area do not normally relocate due to flooding events. In case of severe flooding, some people may temporally evacuate over a short distance, to return to their houses after the flood. When households were asked about permanent relocation, many of them have expressed that "it is very hard to live here but we are born here, our forefather lived here then why should we leave?" They relocate permanently only when they face riverbank erosion. About 49% of households relocated during the entire period 1962–2016, and more than 95% of households claimed that riverbank erosion is the main reason of relocation. As expected, the maximum number of relocations occurred in the char areas, i.e., SHS2 (Figure 5a). By considering the socio-economic status, the maximum number of relocations occurred in poor (76%) and moderately poor (70%) households (Figure 5b). More than 90% of respondents relocated within 5 km from their previous locations and about 40% poor and 30% moderately poor households are considering relocating again because of riverbank erosion. A statistical analysis was performed and it was found that there exists a significant difference in relocation by the respondents between SHS and socio-economic groups (with α = 0.05).

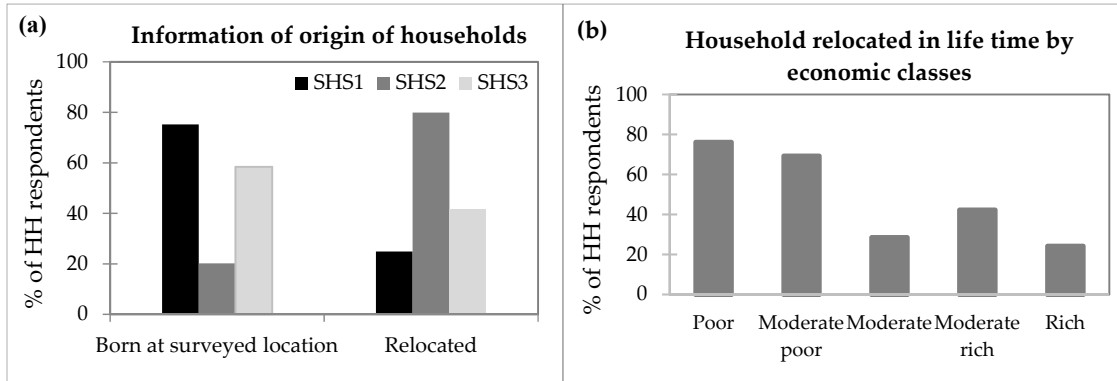

**Figure 5.** Households relocated due to riverbank erosion. (**a**) Information of origin of households. (**b**) Household relocated in lifetime by economic classes.

Only 5% of households mentioned that members had to change their occupation due to flood (Figure 6a). They were mainly from the poor and moderately poor classes (Figure 6b). They changed their occupation mainly from farmer to day laborer, rickshaw puller, and fishermen. The number of households reporting a change in occupation is low because alternative employment opportunities are limited. More than 90% of households mentioned that they have nothing to do during flood events and a scarcity of work arises during those periods. Normally, they look for day laborer work during flood events. A statistical analysis was performed and it showed that there is no significant difference in change in employment by the respondents between the SHS but there exists a significant difference in change in employment by socio-economic groups (with α = 0.05).

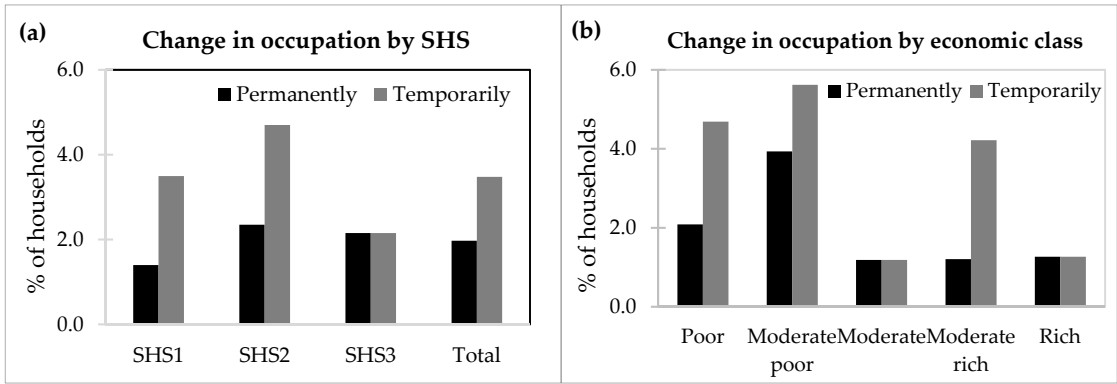

**Figure 6.** Change in occupation of households in the study area. (**a**) Change in occupation by SHS. (**b**) Change in occupation by economic class.

The results show that most households do not change their cropping pattern due to flooding. Using indigenous knowledge, they developed a unique technique to cultivate rice which they follow every year. However, in severe floods this technique does not work and they lose the whole harvest. About 20% of households cultivated fast growing crops after the severe flood in 1988, and about 15% of households did so after the flood events in 2007 and 2015. A few farmers mentioned that they keep their lands fallow during the flood season to avoid losses.

In the study area, inhabitants raise the plinth of their houses to prevent flood water to get into their home. Only 11% of household respondents have also raised their homestead platform. They all are from rich and moderately rich socio-economic classes.

Inhabitants of the study area are undertaking some other adjustment strategies such as storing food for flood event, constructing houses with movable materials, planting trees to avoid erosion. A few of them are also temporarily going to big cities to earn some money as day laborers to overcome the flood losses.

## 4.2. Impacts of Flooding and Riverbank Erosion

By analyzing the negative impacts of flooding and riverbank erosion, it was found that the average income is decreasing over time. About 98% of household experienced decrease in income in every year. Households claim to lose about 90% of their monthly income during severe flood events (Table 3). A statistical analysis was performed and it showed that there is no significant difference in income loss of the respondents between the SHS and by socio-economic groups (with $\alpha = 0.05$).

**Table 3.** Decrease of income due to floods in the study area.

| Parameters | Socio-Hydrological Spaces | | | Socio-Economic Classes | | | | | Whole Study Area |
|---|---|---|---|---|---|---|---|---|---|
| | SHS1 | SHS2 | SHS3 | Poor | Moderate Poor | Moderate | Moderate Rich | Rich | |
| % of households experiencing a decrease of income | 97% | 99% | 96% | 97% | 98% | 97% | 97% | 99% | 98% |
| % of monthly income lost during severe floods | 86% | 88% | 91% | 88% | 88% | 89% | 88% | 90% | 89% |

Moreover, 71% of households face an increase in monthly expenses in times of floods (Table 4). Such an increase is about 60% of the monthly income with little (but not significant, see ESM) differences between SHS and by socio-economic groups (Table 4). Decrease in income loss and increase in monthly expenses due to flood and riverbank erosion are generating the indebted situation for the households in the long run.

**Table 4.** Increase in expenses due to flood in the study area.

| Parameters | Socio-Hydrological Spaces | | | | Socio-Economic Classes | | | | Whole Study Area |
|---|---|---|---|---|---|---|---|---|---|
| | SHS1 | SHS2 | SHS3 | Poor | Moderate Poor | Moderate | Moderate Rich | Rich | |
| % of households facing an increase in expenses in time of flood | 71% | 77% | 66% | 78% | 67% | 69% | 71% | 70% | 71% |
| % of monthly income of the average increase of expenses | 63% | 62% | 60% | 66% | 50% | 61% | 52% | 78% | 62% |

### 4.3. Impoverishment

Land loss caused by river erosion is the major loss for the inhabitants of the study area. Land loss data were used to better analyze whether impoverishment differs by SHS and socio-economic classes. On average, 0.9 ha of land per household was lost between 1962 and 2016. As expected, the inhabitants of SHS2 had the highest losses with 1.4 ha per household (Figure 7a). About 10% of households lost all their land since 1962 and 20% of households lost more than 80% of their land. Analyzed by socio-economic status, it is found that the poor households have lost the most, i.e., 1.5 ha per household (Figure 7b). Moreover, poor and moderately poor households lost the highest proportion of their land, many ending up with no land at all (Figure 7c,d). The households of SHS2 have lost about 80% of their lands during this period. A statistical analysis was performed and it showed that these differences in land loss between the SHS and socio-economic groups are significant (with $\alpha = 0.05$).

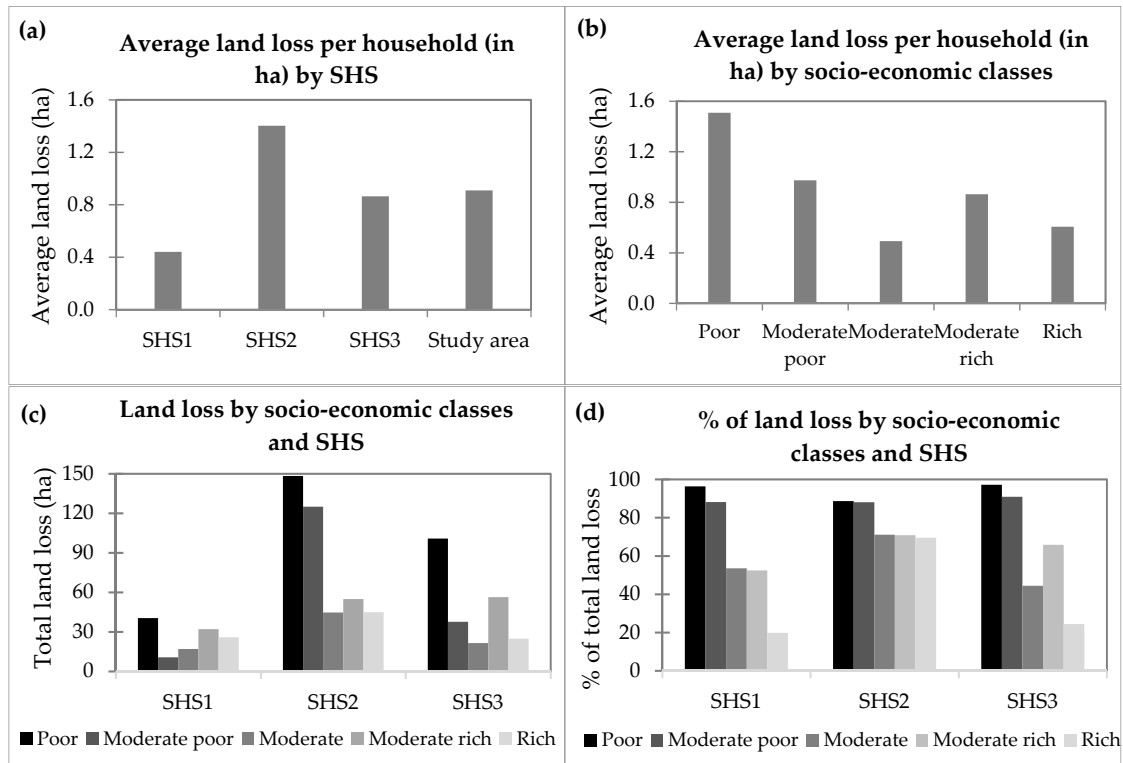

**Figure 7.** Total land loss by SHS and economic classes in the study area from 1962 to 2016. (**a**) Average land loss per household (in ha) by SHS; (**b**) Average land loss per household (in ha) by socio-economic classes; (**c**) Land loss by socio-economic classes and SHS; (**d**) % of land loss by socio-economic classes and SHS.

The survey data also include time series on agricultural crop loss, homestead loss, and other asset losses caused by flooding and riverbank erosion. In terms of total asset losses, households in SHS3 have lost the most, around 2000 USD per year, while the households in SHS2 have lost the least, around

1250 USD per year. Households in SHS2 are better prepared, since they live in the char area, and have less to lose because they are, on average, poorer. By analyzing asset losses by socio-economic status, one can see that poor households have lost most assets (more than 2000 USD). Most poor people today claimed to be richer back in the 1960s, but that they became poorer because of repeated asset losses caused by flooding the riverine erosion. Yet, it should be mentioned that no significant correlations between flood severities and reported losses were found (details are given in ESM).

The respondents were asked about their recovery processes to explore how they recover from losses caused by consecutive events. "Flood and riverbank erosion have snatched everything from us, and we become destitute", is one of the most commonly heard expressions in the char area (SHS2). People in the areas also like to ease their situation by believing that "Almighty Allah (God) has sent this flood towards us and so we have to accept it. This is our fate". Most respondents informed us that it is almost impossible to recover from the riverbank erosion. About 53% of the respondents mentioned that they could not recover at all from the combined losses of flooding and riverbank erosion (Figure 8a). Only 5% of the households recovered a little but not enough to get back to their previous position. They recovered partially with hard working as agricultural laborer, other day laborer employment, and by selling properties or taking loans to do business, cattle farming, or send a household member to temporarily migrate to cities to earn some money. Also, more than 80% of the households in SHS2 could not recover at all, which is highest among the SHS. Looking at socio-economic classes, one can see that more than 70% of poor and moderately poor households could not recover at all (Figure 8b). A statistical analysis was performed and it showed that there is no significant difference in recovery from loss by the respondents between the SHS but there exists a significant difference in recovery of loss by socio-economic groups (with $\alpha = 0.05$).

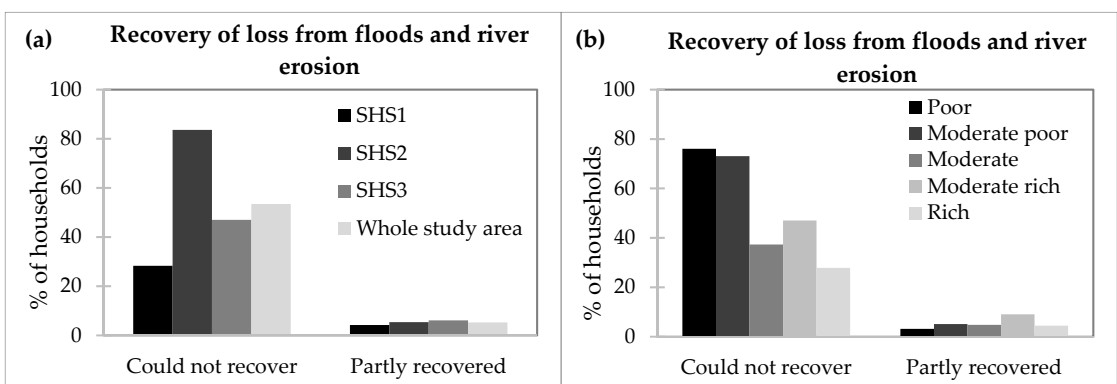

**Figure 8.** Recovery of loss from floods and riverbank erosion in the study area. (**a**) Recovery of loss from floods and river erosion; (**b**) Recovery of loss from floods and river erosion.

These statistics do not tell the whole story, since they present averages. During the field survey a few exceptions were found to the overall picture sketched above. While most of the people suffer, a few rich households benefited from flooding and riverbank erosion by giving loans with very high interest to the flood victims. Statistics do not show the richness of individual experiences either, where each household has its own story to tell such as the one presented in Box 1. The situation of Mr Abdul Baki Khan is typical for some char dwellers, but the number of times his family moved is also quite exceptional compared to other households.

Box 1. An example of a life course of impoverishment in the study area.

**Story of Mr Abdul Baki Khan, an example of living with floods in the study area**

   Mr Khan, who is 63 years of age, lives in a char of the Jamuna River in Uria union of Fulchari upazila of Gaibandha district. He is a landless farmer whose household consists of seven members. They organize their livelihoods around the river. They grow maize, jute and rice on a field by the Jamuna River char. The occurrence of regular flooding events, associated with eroding land, forces them to relocate every two to three years. Over the period 1978–2018, the family moved 13 times. Because of growing population in the area, finding possible places to live and farm has become increasingly difficult. In the past, he used to have access to some 9 hectares of land (large famer according to the agricultural land size), but today he has become landless. Now he is working on the lands of others and his eldest son is also working with him. His income from agricultural land is often not enough for the family, so they limit their daily needs. From morning to afternoon, he and his son go to the field and work hard. Despite these efforts, their monthly income is only about 50 USD. Their house is made of locally available materials with earthen floor, wood, paddy straw with tin on the top of the house. Mr Khan is an example of many people living in the Jamuna floodplain and its char. Many people like him used to be richer earlier, but they have become poorer because of flooding and erosion.

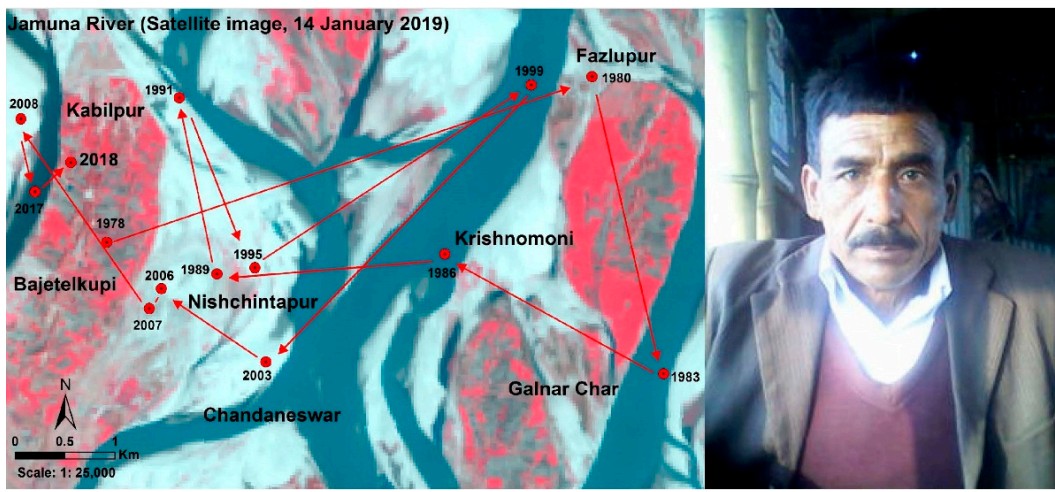

All relocations of Mr Khan and his family mapped onto the current Jamuna River pattern.

## 5. Discussion and Conclusions

   Primary and secondary data were analyzed to reveal how the three SHS and the socio-economic classes are affected by flooding and bank erosion, and their inhabitants adopt adjustment measures to face these hazards and reduce or mitigate the related risks. This study quantified the costs of living with floods and showed that this can lead to general impoverishment in the long term.

   While the country (Bangladesh as a whole) has become more prosperous [14], the per capita gross domestic product (GDP) of Bangladesh is accelerating rapidly [49], and the percentage of the population living below the national poverty line is decreasing [50]. Yet, the prosperity of the study area is below the national level, partly because growing GDP in Bangladesh is mainly urban [39]. According to the National Accounts Statistics for the year 2016 [51], the per capita annual income of the study area is estimated at 1160 USD compared with the national average of 1544 USD. Based on the surveyed data, it is estimated that per capita annual income in SHS1 at USD 1200, in SHS2 at USD 700, and in SHS3 at USD 1000 (below BBS estimates). It is also estimated that the percentages of households below poverty line is about 18% in SHS1, 32% in SHS2 and 15% in SHS3. Given that the main source of income in this area is represented by farming [34], the occurrence of flooding in the area highly impacts income generation. In SHS2, inhabitants only cultivate one crop per year and experience flooding essentially every single year. In SHS1 and SHS3, flooding occurs roughly once every two years. In all three SHSs, over 90% of the respondents experienced over the years an average income loss of about 80 USD which they attribute to flooding, which corresponds to about 85% of their monthly income. In addition, riverbank erosion is responsible for loss of land and assets in the area.

According to the surveys, households from all three SHS and all socio-economic classes have lost land over the past 50 years. SHS2 people have lost up to 80% of their land and many households in the poorest socio-economic class stated that they have lost all their land. Inhabitants that relocated several times became poorer more quickly than those who did not need to relocate as frequently. Bank erosion victims tend to relocate to nearby land, hoping that they can recuperate their land in the near future. There are no many alternatives to recover from income, land, and asset loss. They either change their occupation from farmer to day laborer, fisherman, or temporarily migrate to cities to earn. However, recovery rate is very low: income loss and asset loss recovery is marginal even for rich class people. Furthermore, due to the lack of employment alternatives in the area, many people need to take loans with high interest during the flood season. In an attempt to increase their loss recovery capacity, people try to save some money for the flood season; however, their savings evaporate rapidly during the flood season when monthly expenses can be higher than incomes. As a result, they move towards impoverishment over time. Household respondents are affected by flooding and riverbank erosion in terms of income losses (Table 3), increases in expenses (Table 4) as well as land losses (Figure 7). The household respondents have also mentioned that it is almost impossible to fully recover the loss (Figure 8). This study analysis showed that although flood mortality rates in Bangladesh have been significantly decreasing over time [52,53], the costs of adjusting to flooding and riverine erosion are very high and can negatively impact the livelihood of local people. While households mainly blame flooding and bank erosion, there can be other socio-economic factors, such as a lack of investments in alternative types of employment, which also can significantly contribute to the impoverishment of the area.

Figure 9 depicts the definitions of coping and adapting introduced in Section 1. This research shows that households in the study area are coping (but not adapting) with flooding and riverbank erosion, since the net result of their adjustments is a decline in socio-economic conditions (see orange trajectory in Figure 9).

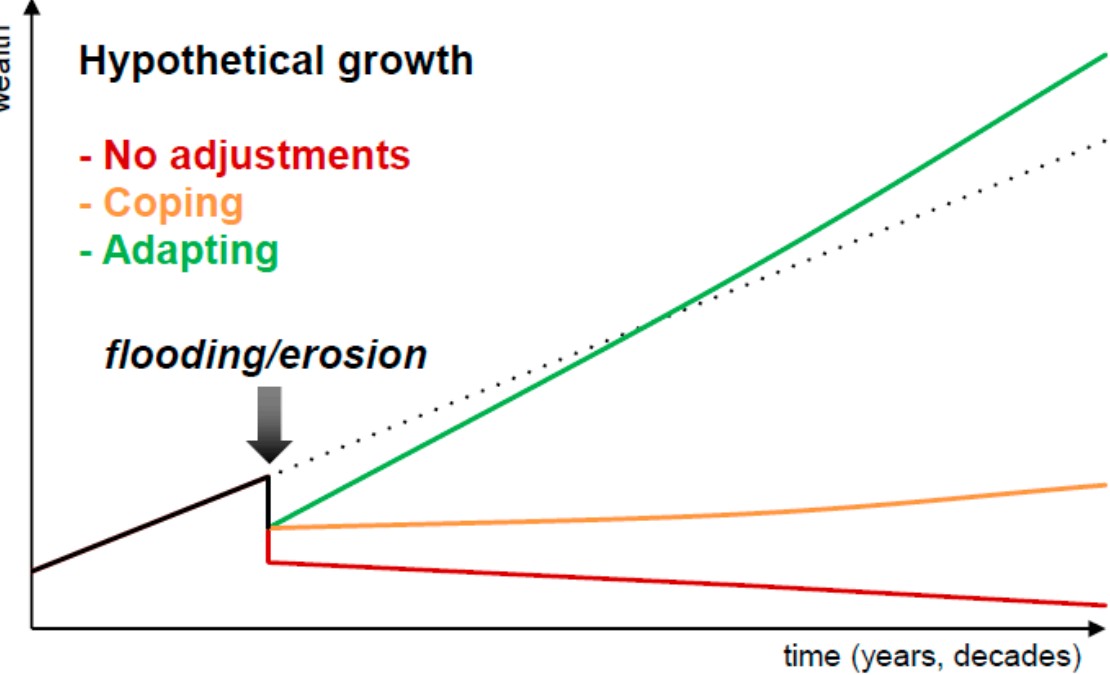

**Figure 9.** Coping or adapting? Deviation from a hypothetical trajectory of socio-economic growth (black line) caused by flooding and erosion, in case of: (i) no adjustments (red line), (ii) coping (orange line), and (iii) adapting (green line).

Loss of homesteads forces people to move to new places without any option and puts them in desperate situations. Despite these extreme living conditions, the char inhabitants do not leave this area

because of various obstacles, the lack of available land elsewhere being foremost. They try to stay as long as possible in their home during high floods and in the case of erosion, dismantle it and transport it to another char which is less or not affected by erosion at that moment. While the situation in the char area (SHS2) is extreme, in every location of the study area people are losing income and assets, since they can only partially recover from flood events. Overall, they are getting poorer and therefore less able to make the further adjustments for the next flooding event. Although there are most likely other factors at play, repeated flooding and associated bank erosion contribute to overall impoverishment (in relative terms compared with national figures of economic growth) of the floodplain people of Bangladesh, despite their use of multiple strategies to respond to flooding and riverbank erosion. Due to the scale of works, the flood control measures constructed by the Government are not likely to ever be sufficient to prevent flooding or riverbank erosion. This research posits that indigenous adjustments, such as the adoption of different types of crops to varied flood depths, should be reinforced to reduce the negative impacts of flooding on agriculture and therefore slow down the rate of impoverishment in the area.

This research work does not only contribute to advance the knowledge about socio-hydrological dynamics in Bangladesh, but also provides more general insights for flood risk management in low-income regions of the world. There is an ongoing discussion about the need for a shift from hard (fighting floods) to soft (living with floods) approaches [54]. It has been argued that low-income countries such as Bangladesh should not implement hard engineering work to increase levels of structural flood protection, as it has been done in most Western countries, but stick to their traditional softer approach of living with floods [55]. Indeed, some of the polders that were constructed in 1970s had negative impacts on ecosystems and livelihoods and are currently being revised to re-establish a workable sediment and water balance [56]. However, the Jamuna case study demonstrates that living with floods has enormous costs and it prevents socio-economic growth in the areas. As such, there are no clear-cut answers to the question of how low-income countries should deal with flooding. This research argues that universal recipes do not exist. Instead, there is a need to find trade-offs between hard and soft options depending on the values given by local communities, experts, practitioners, and governments to environmental, social, and economic benefits and costs of alternative strategies. A better understanding of socio-hydrological dynamics, such as the one provided with the Jamuna case study, can help identify these trade-offs by shading light on both the positive and negative effects of living with floods.

**Supplementary Materials:** The following are available online at http://www.mdpi.com/2073-4441/11/6/1238/s1, Document S1: ESM Living with flooding in the Jamuna floodplain in Bangladesh: limits to adaptation, Table S1: EMS Literature review: Living with flooding in the Jamuna floodplain in Bangladesh: limits to adaptation.

**Author Contributions:** Conceptualization: M.R.F., A.W., L.B., K.S., M.Z. and G.D.B.; Data curation: M.R.F.; Formal analysis: M.R.F., A.W., L.B. and K.S.; Investigation: M.R.F., A.W., L.B. and K.S.; Methodology: M.R.F., A.W., L.B. and K.S.; Supervision: A.W., L.B., M.Z. and G.D.B.; Writing—original draft preparation: M.R.F., A.W. and L.B.; Writing—review and editing: M.R.F., A.W., L.B., M.Z. and G.D.B.

**Funding:** The research is funded by NWO-WOTRO grant W 07.69.110 "Hydro-Social Deltas: Understanding flows of water and people to improve policies and strategies for disaster risk reduction and sustainable development of delta areas in the Netherlands and Bangladesh". Giuliano Di Baldassarre is also supported by the European Research Council (ERC) within the project "HydroSocialExtremes: Uncovering the Mutual Shaping of Hydrological Extremes and Society", ERC Consolidator Grant No. 771678. Luigia Brandimarte is supported by the Swedish Strategic research programme StandUP for Energy.

**Acknowledgments:** Special thanks to Md Mahabubar Rahman, Md Enamul Haque, and Md Shahrier Islam for collecting household data from the study area. Special thanks to Abdul Baki Khan for allowing us to use his photo and life story in this paper. Special thanks to Engr. Md Waji Ullah, Executive Director, CEGIS for providing satellite images of Jamuna River for the study.

**Conflicts of Interest:** The authors declare no conflict of interest.

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
