# Peer review of "The Costs of Living with Floods in the Jamuna Floodplain in Bangladesh"

_water, doi:10.3390/w11061238_

Round 1
Reviewer 1 Report
The paper by Ferdous et al. examines the long term economic impact of flooding and adaptation strategies in the Jamuna floodplain in Bangladesh. The paper takes a novel and useful approach of quantifying the impacts of flooding and adaptation to flooding from the bottom up by conducting and analyzing a field survey of residents across the region. The paper is well written, interesting, and potentially valuable. There are some minor issues that should be addressed prior to publication and a few suggestions.
The abstract is too long and meanders a bit. I suggest cutting out the discussion on what adaptation is and sequencing the abstract along the lines of:
1. Residents of the Jamuna floodplain use adaptation to successfully mitigate flood risk, but,
2. Adaptation has unintended consequences that, over the long term drive residents towards poverty.
3. Here we conduct a survey to quantify...
4. We find...
The introduction is a bit confusing as organized:
1. background information on flooding and coping
2. introduction to the purpose of the paper
3. discussion on the history and some prior research
4. 80,000 ft view of the methodology and summary of findings.
Consider reordering to 1,3,2,4.
Throughout there are minor stylistic issues and grammar errors that impede clarity that should be addressed. As examples:
"In this paper we endeavor to shed light on" would be more clearly conveyed by "In this paper examine the..."
" Only a few households have recovered a little but not enough to get back to their previous position: 5% of households recovered partially with hard working as agricultural labourer, other day labourer employment, and by selling properties or taking loans to do business, cattle farming, or send a household member to temporarily migrate to cities to earn some money etc." Is a long sentence that is difficult to parse and understand.
" There are no many alternatives to recover from income, land and asset loss: change of activity (farmers will work as day labourer or turn into fisherman) or migration towards the city (mainly temporal migration to earn money to recover)."
" We have showed that household respondents are facing income loss (Table 3) followed by crop loss and land loss due to flooding and riverbank erosion (Figure 7)." Is more simply stated in present activce voice as, "We show..."
" The situation of Mr. Abdul Baki Khan is typical for some char dwellers (SHS2), but the number of times his family moved is also quite extreme compared with others in SHS." is unclear. is it extreme compared to households in other SHS's?
" We get this information in field in an informal composition and thus we don’t have statistics but it came from discussions with the household respondents." is unclear.
Instances of awkward or extraneous wording such as "Embankmnets sometimes even worsen..." which could be rewritten "Embankments sometimes worsen...".
I remain a bit confused as to the overall trend of wealth in Bangladesh. Section 6 discussion notes that the country has become more prosperous, but text in section 3.2 seems to indicate the opposite. The article shows the strong relationship between extreme flooding and long-term economic decline in the study area. If GDP in increasing nationwide it seems like this would strengthen the paper's conclusions.
The results can be difficult to follow, they are not presented consistently. For instance, income loss is presented using:
1. percentage of households that reported a decrease of income every year (which is clear as well as astounding).
2. monthly income loss due to flooding in dollars (but it is not clear over what timespan this is reported, i.e. total over the entire survey time-span, per flooding event, or per year).
3. ratio of income loss to income in severe flood events.
Then increases in expenses are presented using:
1. percentage of households facing an increase in monthly expenses in time of flood.
2. average increase in monthly expenses in time of flood.
3. average increase as a percentage of monthly income
Then land loss is presented as:
1. average acreage lost over the time period.
2. percentage of households that lost all land.
3. percentage of lost land over the time period.
It is hard to keep track of the aggregate picture painted by these statistics.
Other suggestions:
It would be informative to have the survey questions that were used to develop each measure.
Quotes from the survey such as "‘Flood and riverbank erosion have snatched everything from us and we become destitute,’ go a long way towards putting the statistics in context, and I would like to see more of these.
I found that visualizing the statistics was difficult. The nature of the data suggests a plot with time on the x-axis and land or total wealth on the y-axis to show the trends over time.
One or more typical case studies examined and presented as a timeline might be effective, perhaps in the supplemental material.
The shifting of past, present, and future tense use can be distracting. For instance, in the last paragraph of the introduction you use both past and present tense to refer to your research for this article, and future tense to refer to topics addressed later in the paper. Consider using present tense for all discussion of the research and location of information in the paper. Similarly, there are places where you discuss Bangladeshi response to flooding in present (e.g. Section 3.1 homesteads) future (e.g. Section 3.1 survival) and past (e.g. Section 3.1 agriculture) tense. Recommend you stick with present tense, and carefully check through the article for consistency in tense.
Author Response
Thank you very much for the positive and nice comments to our work, and for providing constructive comments and suggestions that helped us improving the description of our work. We carefully addressed all points as described in our point-by-point response.

Reviewer 2 Report
See the attached file.

Author Response

(The authors gave the same response as above.)

Reviewer 3 Report
This manuscript describes how households have adapted over recent decades to flooding and river bank erosion in a region of Bangladesh. The novelty of the paper is the use of quantitative household data that spans many decades. This research challenges the prevailing positive assumption that “Bangladeshis know ... how to respond” to climate change with many households merely coping (line 43). It is an interesting manuscript, but would benefit from some tightening of the English and there is repetition (e.g. regarding relocation).
I think that the major limitation of the paper is the discussion and conclusions. Quite detailed information has been collected, but how is it to be used, for example with regards to successful adaptation. Line 85: “We are interested in finding out which households succeed in adapting” and line 98 “we hypothesize that differences in socio-economic status ... matter for households ability to adapt”– these statements should be revisited and clear statements should be given. Are there any success stories or is adaptation simply not possible (which seems to be the case for SHS2). Line 120 states the paper “conclude with brief discussion of policy implications”, this is not evident.
Minor edits:
Line 339: Does selecting household headed by older men introduce any selection bias to the analysis?
Line 428: Check this sentence, the decrease in monthly income is lowest for moderately poor at 75USD. Are all figures adjusted to a reference year?
Line 462: states total asset loses of 1,250USD per year; given the income data in Table 2 this appears very large, I assume this figure is because of land loss. How large are households?
Figure 9: Not really required, although placing in a national context is appropriate.
Line 562: “We found that .. . poor people have less to lose to start with”. I found this a rather strange statement and would reword.
General question: How is land divided between relatives (inheritance), is it divided or passed to an individual, does this contribute to changing farm size.
Author Response

(The authors gave the same response as above.)

Reviewer 4 Report
Abstract:
Successful adaptation means that individuals, households and societies make adjustments to their lives and livelihoods while benefiting from opportunities that come with extreme climate events such as flooding.
"In this paper, we show how the adjustments to environmental conditions can also have negative outcomes..." and this why the sustainability of adaptation measures needs to be assessed for example based on Adger (2005) criteria.
.".., and show that residents mainly cope with the situation for the time being, and do not achieve successful adaptation." Because they usually adapt reactively, based on what is available for them currently. So planned adaptation is not on the table for Bangladesh.
Line 39 to 48: This is a very big quote. It needs to be rephrased properly. This considered as plagiarism despite the quotation marks.
... Bangladesh Confronts Climate Change: Keeping Our Heads Above Water” (write the title in Italics)
Bangladeshis are good "adapters". Adapters is not a good word to use. You could use for example Bangladeshis have a good adaptive capacity or less vulnerable to the negative consequences of climate change effects. So please do not use the word adapters in your text anymore.
"How have generations of living with floods affected them? Are they indeed good adapters, or are they
merely coping at the expense of? " Questions are not well written or understood.
Line 77: "... In the worse flood in living memory, ... " The sentence is not well-written.
"Casualty rates" should be causality rates.
You can use maladptation to refer to impoverishment. In fact, the human agent is now at the center of the adaptation process. This paradigm shift can lead to maladaptation.
Line 115 to 117:
"We were able to do a long term analysis of socio-economic development by making use of our unique dataset of approx. 900 households in an area along 30 km of the Jamuna River in the north of Bangladesh (described in detail in Section 4)." How does this make a 'long-term' analysis; I do not understand.
Line 532-535: In this case, I do not see how people in the case study are adaptation well to flooding, especially in the long-term.
What about residual impacts that come the effects of flooding?
Author Response

(The authors gave the same response as above.)

Round 2
Reviewer 2 Report
The authors greatly improved the paper and handled the comments mostly satisfactorily. One major point and several smaller points.
The point on a more general outlook should, I think, not be easily discarded since the audience is international and interdisciplinary (see the aim of the journal). Especially the very pragmatic reason that the authors aim to collect more cases to compare is not satisfying. I think it is both common practice ánd good scientific conduct to position single case studies and their lessons in the wider field. There are general implications which need to be stipulated upon, despite the particularities of a case. Including the general outline will strengthen the case selection reasons in the methods section and the overall conclusions, which are now somewhat superfluous and more a continuation of the results section than a proper discussion of the findings and conclusions. To cite for example nature.com: “The Conclusion section presents the outcome of the work by interpreting the findings at a higher level of abstraction than the Discussion and by relating these findings to the motivation stated in the Introduction.” And “show what your findings mean to readers.”
Several other, smaller issues: the limitations are now mentioned at the end of the results. This should be either in the methods sections (they are methodical choices) or in the discussion.
Although the authors looked at the tense, spelling and language in general this is still off, also due to the revisions (for example l636 he à the), l727-728: “The extreme situation is the char area” needs a word after ‘is’: shown, in (but the whole sentence is quite strange and might be altered). Please do an additional language check.
L730-l734: these are strange additions in a discussion/conclusion section since they are data excerpts. Quotes from interviews or focus groups are normally only used in the results section. (also relates to my first point about the discussion/conclusion section)
Figure 9 does not match the text: there is no decline in the orange -coping- line, but a slower increase compared to adapting.
Please delete some references to the ESM, not each statistical notion needs to be accompanied by (details are in the ESM).
Author Response
We thank the Reviewer for the positive and nice comments to our revision work, and for providing additional constructive comments and suggestions that helped us further improve the description of our work. We carefully addressed all the remaining points as described in our point-by-point response.

Round 3
Reviewer 2 Report
The authors did sufficiently improve the paper based on the comments, although the quality of the paper might benefit from further work on the general outlook and contribution to the scientific field of flood risk management in both the introduction and discussion/conclusion sections.